**∂ | Open Peer Review** | Clinical Microbiology | Research Article

# Clinical laboratory evaluation of the Hologic Panther Aptima BV and CV/TV assays for the diagnosis of vaginitis in Dunedin, Aotearoa New Zealand

Juliet Elvy,[1] Katelyn Carter,[1] Jenna Paterson,[1] Megan Smith,[1] Gayleen Parslow,[1] James E. Ussher[1,2]

**ABSTRACT**    Vaginitis presentations are common, but traditional diagnostic methods are imperfect. Molecular methods for bacterial vaginosis (BV) and vulvovaginal candidiasis (CV) are increasingly available but not commonly utilized in Aotearoa New Zealand. We evaluated the Hologic Aptima BV and CV/*Trichomonas vaginalis* (TV) assays against our current methods (Gram stain, yeast culture, and Hologic Aptima TV assay) and performed a retrospective BV clinical audit. The BV Aptima assay performed well with high sensitivity (97.5%) and specificity (96.3%) when the indeterminate BV category was excluded. BV indeterminate samples were almost evenly split between positive and negative results when tested on the Aptima BV assay. BV Gram stain interpretation was error prone, with 20% of samples discordant on duplicate examination. Although the Aptima CV assay was highly sensitive, it lacked specificity compared with Gram stain (83.5%) but was similar to culture (91.2%). Our BV clinical audit showed that patients with a BV indeterminate result were less likely to be treated for BV than those with a positive result, meaning more women may be treated for BV if this assay were implemented. Overall, implementation may improve laboratory workflow and consistency of reporting, but cost may be a barrier. The clinical impact of changing methods needs to be considered.

**IMPORTANCE**  In this paper, we evaluate the performance of the Aptima molecular assays against current Gram stain and culture methods, as well as a clinical audit to determine the potential clinical impact of implementation. Although molecular methods are increasingly used in other countries, New Zealand has not yet adopted this approach. Importantly, we found Gram stain for bacterial vaginosis (BV) to be error prone, with 20% of Gram stain results discordant on repeat examination. We show the potential for molecular methods to increase BV diagnoses and improve reproducibility and consistency of reporting which, according to our clinical audit results, would lead to more women being treated for this dysbiosis condition overall.

**KEYWORDS**  vaginitis, bacterial vaginosis, vulvovaginal candidiasis, *Trichomonas vaginalis*, Nugent score, Gram stain

Vaginitis presentations comprise a significant number of healthcare consultation visits, with bacterial vaginosis (BV), vulvovaginal candidiasis (CV), and *Trichomonas vaginalis* (TV) as leading causes (1).

BV is a vaginal dysbiosis condition, characterized by a change in the normal vaginal flora from *Lactobacillus* species to anaerobic pathogens like *Gardnerella vaginalis*, *Mobiluncus* spp., and *Fannyhessea vaginae* (formerly known as *Atopobium vaginae*) (1, 2). Symptoms of BV include offensive vaginal discharge, although up to 50% may be asymptomatic (3). The prevalence of BV is estimated to be up to 25% globally, but this

**Peer Reviewer** Megan Amerson-Brown, University of Alabama at Birmingham Bone Marrow Transplant, Birmingham, Alabama, USA

Address correspondence to Juliet Elvy, juliet.elvy@awanuilabs.co.nz.

The authors declare no conflict of interest.

may vary according to demographic and patient factors (4). The prevalence of BV in Aotearoa New Zealand (NZ) has not been documented.

CV is caused by yeast infection, predominantly with *Candida albicans* (80%–90% of cases) (5). Yeasts are considered part of the normal vaginal flora, but overgrowth and infection can lead to symptoms like abnormal discharge, itch, and irritation. Surveys suggest about 75% of women will develop CV at least once in their lifetime, and 40%–45% will have two or more episodes (5, 6). Although many causative yeast species have undergone recent taxonomic changes (e.g., *Nakaseomyces glabrata*, formerly known as *Candida glabrata*) (7), the term CV will be used throughout this article.

TV is a sexually transmitted protozoon parasite and the third most common cause of infectious vaginitis. TV infection classically presents with profuse, frothy, and offensive vaginal discharge, but asymptomatic infection is also common. Infection during pregnancy may increase the risk of adverse outcomes (8).

There is overlap in the symptoms and signs associated with specific aetiologies of vaginitis, and clinical diagnosis alone is difficult. Laboratory testing to help guide diagnosis and appropriate clinical management is therefore desirable (5). For BV, microscopic examination of a high vaginal swab with Gram stain and Nugent score is considered the gold standard (8–11). Nugent score is a semi-quantitative scoring system, which identifies any change from the normal vaginal flora (typically dominated by Gram-positive lactobacilli) to anaerobic overgrowth (dominated by Gram-negative and Gram-variable BV morphotypes) and returns a positive, negative, or indeterminate categorical result. The BV indeterminate category does not definitively rule in or rule out the diagnosis, and the clinician must decide if treatment is warranted (9, 10). For CV, there is no consensus gold standard, with some guidelines recommending microscopy (to look for neutrophils and budding yeast with pseudohyphae), whereas others advise yeast culture (5, 12, 13). TV has traditionally been diagnosed by wet mount microscopy or culture, but more recently, TV nucleic acid amplification tests have become the recommended diagnostic method available, with improved sensitivity (8).

Our laboratory is located in Ōtepoti/Dunedin in the lower part of Te Waipounamu, the South Island, of NZ. We are fully accredited with International Accreditation New Zealand (IANZ) and the sole provider of laboratory services for both community and hospital patients for a population of around 350,000, stretching over a geographical area of 62,500 km$^2$. We receive on average 1,800 vaginal swabs per month and undertake Gram stain with Nugent score for BV, Gram stain for the presence of yeast cells and pseudohyphae, and yeast culture. We also perform molecular testing for TV, *Chlamydia trachomatis* (CT), and *Neisseria gonorrhoeae* (NG) using the Hologic Panther Aptima system. We have diagnostic stewardship interventions in place whereby clinical details must be clearly stated on the request form. This approach aims to limit unnecessary and clinically inappropriate testing.

Gram stain methods for BV and CV are labor-intensive, require skilled operators, and interpretation is subjective. Recent work has demonstrated the utility of molecular methods to replace the Gram stain, with an increasing number of commercially available molecular assays now available (1, 14–17). Although molecular methods are considered routine in NZ for TV, CT, and NG, this is not the case for BV and CV. We sought to investigate whether the Hologic Panther Aptima BV and CV/TV assays would be suitable for our setting by comparing performance against current methods.

One key difference between the Aptima BV assay and Gram stain with Nugent score is the absence of indeterminate BV results. To better inform the potential clinical impact of implementing this assay, we performed a retrospective clinical audit to document current BV prevalence and treatment rates in our region.

## MATERIALS AND METHODS

### Aptima BV and CV/TV assay evaluation

We utilized anonymized vaginal swab samples submitted to our laboratory for routine diagnostic purposes from April 10th to May 19th, 2023. Samples were included if they were from patients aged between 14 and 60 years and if both an Amies agar gelbacterial swab (Copan Diagnostics Inc, Murrieta, CA) and an Aptima Multitest swab (Hologic Inc, San Diego, CA) were received. Samples were excluded if they were outside the age range or if only one swab type was received. Samples without accompanying clinical details indicating that the patient was symptomatic were handled according to current protocols, that is, not processed, but stored for 7 days in case the relevant clinical information for processing was later provided. Swabs are transported by courier to the testing laboratory. All swabs were held for 7 days following standard testing and reporting before anonymization and inclusion in the study. Study samples were assigned a specific study number with all identifiable information removed prior to processing. No additional clinical reports were produced as a result of the assay evaluation. This assay evaluation was performed in accordance with standard quality improvement procedures as outlined by the New Zealand Health and Disability Ethics Committees (HDEC), the New Zealand Human Tissues Act 2008, and IANZ.

The Aptima BV and CV/TV molecular assays under evaluation were compared against the following reference methods, which were performed on all samples included in the study: (i) consensus Gram stain and Nugent score for BV, (ii) consensus Gram stain and culture on Sabouraud dextrose agar (SAB) for CV, and (iii) Aptima TV assay (Hologic Inc, San Diego, CA). Gram stains were prepared on glass slides from the Amies swab and stained using an Aerospray automated staining device by various trained laboratory staff as part of standard laboratory testing. All Gram stains were read independently by two trained laboratory personnel, with the Nugent score applied as previously described (8). The first reading was reported to the clinician as part of standard laboratory processes. The second reading was performed only after anonymization. Any discordant categorical results for BV between the first and second reading were followed by a third independent read. All Gram stain readings were blinded to any previous result and performed by various competent laboratory personnel. If two of three categorical Nugent scores matched, this was recorded as the final consensus result. Gram stains were also examined twice independently for the presence of yeast cells and pseudohyphae, and concordance was assessed. However, no third reading was performed for discordant categorical results for CV; discordant samples were not included in the comparison analysis. Gram stains were reported as negative if no yeast cells or pseudohyphae were seen, as yeast colonization if there were yeast cells but no pseudohyphae, or as CV if there were pseudohyphae seen with or without yeast cells. SAB plates were incubated at 37°C aerobically for 48 h, and the results were recorded as no growth, scanty growth (less than 10 colonies), or light, moderate, or heavy growth according to whether there was growth of yeast out to the first, second, or third streak on the SAB plate, respectively. Scanty growth was included in the no growth category for the purposes of assay comparison, in line with our current laboratory standard operating procedures. Identification of yeast was based on typical colonial morphology and wet prep where required. MALDI-TOF identification was not routinely performed unless recurrent or refractory CV was noted on the request form, also in line with our standard operating procedures.

Each Aptima swab was loaded directly onto the Hologic Panther system, which is a fully automated random access analyzer. Testing was performed according to the manufacturer's instructions for use (18, 19). Briefly, the Aptima BV assay comprises an *in vitro* nucleic acid amplification test using real-time transcription-mediated amplification (TMA) to detect and quantify organisms associated with normal vaginal flora (*Lactobacillus gasseri, Lactobacillus crispatus,* and *Lactobacillus jensenii*) and organisms associated with BV (*F. vaginae* and *G. vaginalis*). The assay returns a qualitative result (BV positive or negative) based on a proprietary algorithm analysis of the relative detection of

*Lactobacillus* species, *G. vaginalis*, and *F. vaginae* and does not include a BV indeterminate category. The CV/TV assay utilizes the same TMA methodology with targets for *Candida* spp. (*C. albicans, Candida tropicalis, Candida parapsilosis,* and *Candida dubliniensis*), *N. glabrata*, and *T. vaginalis*.

Standard statistical methods were used to calculate the sensitivity and specificity of each assay compared with the comparator method. Samples with inconclusive reference results and samples with invalid or missing investigational assay results were excluded from the analyses.

## BV clinical audit

Vaginal swab data were extracted from our laboratory information system for patients aged 15–60 years from 1st June 2021 to 1st June 2023. Prescription data for a random selection of 450 patients, comprising 150 patients within each BV cohort (BV positive, BV indeterminate, or BV negative), were accessed via each patient's electronic record to determine whether an antibiotic with efficacy for BV (metronidazole, ornidazole, or clindamycin) was prescribed within 30 days after the vaginal swab was taken. Patients were selected using the randomize tool in Excel. Patients for which the prescription date was not able to be accessed via the electronic record portal were excluded. The overall proportion of Gram stain with Nugent score results returned as positive, negative, and indeterminate for the 2-year period was determined, along with rates of BV treatment for the 150 randomly selected patients within each group. Ethical approval was obtained via University of Otago Ethics Committee University of Otago Ethics Committee (reference number HD23/065).

## RESULTS

Three hundred specimens were included in the evaluation. Samples were excluded from the analysis if a consensus BV Gram stain with Nugent score was not available or if invalid (failed). Aptima results were obtained. One sample did not have a yeast culture performed. In total, 285 results were available for both reference and test methods for BV, 271 samples for CV Gram stain, 296 for CV culture, and 287 for TV.

Most patients were NZ European or European ethnicity (188/300, 63%). Ethnicity was not stated for 79 (26%), whereas 18 (6%) were Māori, nine (3%) were Asian, three (1%) were Pacific, and three (1%) were of other ethnicity. The average age was 30 years (range: 15–58 years).

The most commonly reported symptoms on the sample request form were vaginal discharge (218, 73%), itch (67, 22%), pain (46, 15%), and abnormal vaginal bleeding (23, 8%). Sixteen (5%) samples were processed as asymptomatic screening prior to the termination of pregnancy, and 17 patients (6%) were noted to be pregnant.

Using reference methods, 116 (40.4%) patients were negative for BV, CV, and TV. BV was detected in 78 (27.2%) by consensus Gram stain with Nugent score, and CV was detected in 35 (12.9%) by Gram stain or 81 (27.3%) by culture. Three samples (1%) were positive for TV (see Table 1). Infection rates for BV and CV were 38.2% and 32.8%, respectively, for the Aptima assays. Overall, 15.4% of patients with BV by reference method also had CT (12/78). No patients had NG detected. Coinfection rates for BV and CV were 3.3% (9/271, Gram stain) or 6.8% (20/296, culture) by reference method and 12.8% (38/296) by the Aptima assay (see Table 1).

## Aptima BV assay comparison

Consensus BV Gram stain with Nugent score result was available for 287 samples since eight samples had three different categorical Nugent scores, two samples were discordant but did not get a third read, and three samples had only one read performed. Of 297 samples with two reads, 237 (79.8%) demonstrated concordance in the categorical result between the first and second Nugent score, of which 151 (63.7%) were reported as BV negative, 61 (25.7%) as BV positive, and 25 (10.5%) as BV indeterminate.

**TABLE 1** Infection rates by reference method and Aptima investigational assay

| Infection | Result (n, [%]) | |
|---|---|---|
| | Reference method | Aptima |
| BV | 79/285 (27.7) | 109/285 (38.2) |
| CV | 35/271 (12.9) | 97/296 (32.8) |
| – Consensus Gram stain | 81/296 (27.3) | |
| – Culture | | |
| TV | 3/287 (1) | 3/287 (1) |
| CT | 12/296 (4) | - |
| NG | 0 | - |
| BV + CV | 9/271 (3.3) | 38/296 (12.8) |
| – Consensus Gram stain | 20/296 (6.8) | |
| – Culture | | |
| BV + CV + TV | 0 | 0 |
| BV + TV | 1/285 (0.4) | 3/285 (1.1) |
| BV + CT | 11/285 (3.9) | 11/296 (3.7) |
| CV + TV | 0 | 0 |

Sixty samples (20%) demonstrated discordant categorical results between the first and second Nugent scores (see Table 2).

Comparing the BV Gram stain with the Aptima BV assay, 285 comparable results were available. Of these, 160 (56.1%) returned a negative Nugent score, 79 (27.7%) were BV positive, and 46 (16.1%) were BV indeterminate. After excluding samples with an indeterminate BV Gram stain result ($n = 46$), the sensitivity and specificity of the Aptima BV assay were 97.5% and 96.3%, respectively. Of the 46 indeterminate BV Gram stain samples, 26 (56.5%) returned a positive BV Aptima result, and 20 (43.5%) were negative (see Tables 3 and 4).

## Aptima CV/TV assay comparison

Of the 297 samples with two CV Gram stain readings available, 273 (91.9%) were concordant; of these, 234 (85.7%) samples were reported as negative, 35 (12.8%) were reported as yeast infection, and four (1.5%) reported as yeast colonization. Twenty-four samples (8%) yielded discordant results (see Table 5).

Comparing concordant CV Gram stains with culture, one sample negative on Gram stain did not undergo yeast culture, and 35/233 (15%) samples demonstrated growth of yeast despite no yeast being seen on the Gram stain. Of these, 14 (40%) demonstrated moderate or heavy growth. In addition, three of four (75%) samples reported as yeast colonization by Gram stain, demonstrated heavy growth on culture. All samples reported by Gram stain as yeast infection were culture positive, with all demonstrating moderate-to-heavy growth (see Table 6).

Two hundred and seventy-one paired results were available for the CV Gram stain and Aptima CV comparison, since two samples returned an invalid Aptima CV result. Yeast colonization by the reference method was included as negative for CV, yielding a sensitivity of 100% and a specificity of 83.5% for the Aptima assay.

Two hundred and ninety-six paired results were available for comparison of Aptima CV with culture, since one sample was not cultured, and there were three invalid Aptima

**TABLE 2** Concordance of BV Gram stain readings

| First read | Second read | | | | Third read | | | |
|---|---|---|---|---|---|---|---|---|
| | BV negative | BV indet | BV positive | Not performed | BV negative | BV indet | BV positive | Not performed |
| BV negative | 151 | 16 | 4 | 2 | 4 | 10 | 5 | 1 |
| BV indeterminate | 11 | 25 | 16 | 0 | 3 | 12 | 9 | 3 |
| BV positive | 4 | 9 | 61 | 1 | 1 | 4 | 8 | 0 |

**TABLE 3** Comparison of Aptima BV assay against the reference method (Gram stain and consensus Nugent score)

|  | Positive Nugent | Indeterminate Nugent | Negative Nugent | Total |
|---|---|---|---|---|
| Aptima BV positive | 77 | 26 | 6 | 83 |
| Aptima BV negative | 2 | 20 | 154 | 156 |
| Total | 79 | 46 | 160 | - |

results. Scanty growth of yeast was included in the culture-negative category. Compared with culture, the Aptima CV demonstrated a sensitivity of 96.3% and a specificity 91.2% (see Table 4).

Comparing the TV component of the CV/TV assay with the reference Aptima *T. vaginalis*, 13 samples were excluded either due to invalid results or not being tested on both the TV and CV/TV concurrently, with three positive detections and 100% concordance.

## Retrospective BV clinical audit

From 1st September 2021 to 30th August 2023, 25,025 vaginal swab Gram stain Nugent score results were available from our laboratory information system, of which 14,290 (57.1%) were reported as BV negative, 6,266 (25%) as BV positive and 4,469 (17.9%) as BV indeterminate.

Of 150 randomly selected patients within each BV category, three patients per category did not have prescription records available via the electronic record portal and were excluded, leaving 147 per category for analysis. Of those with a positive BV result, 97/147 (66%) received treatment with metronidazole, ornidazole, or clindamycin within 30 days of the result, compared with 38/147 (25%) of patients indeterminate for BV and 7/147 (5%) who were BV negative.

## DISCUSSION

To our knowledge, this is the first clinical evaluation of the Hologic Panther Aptima BV and CV/TV assays in NZ. We found that the BV and TV Aptima assays performed well against our reference methods, with high sensitivity and specificity for BV when the indeterminate category was excluded. BV indeterminate samples were almost evenly split between positive and negative results when tested on the Aptima BV assay. There is a growing number of genera implicated in the BV disease process, which are not detected by the Aptima assay (e.g., *Mobiluncus, Prevotella, and Bacteroides)* (20); future studies could investigate this further by testing indeterminant BV samples with PCR assays, which include more targets, or by metagenomic assessment. Our retrospective clinical audit showed that patients with a BV indeterminate result were more likely to be treated for BV than those with a negative Nugent score but less likely than those with a positive result. Although the Aptima CV assay was highly sensitive, it lacked specificity compared with the Gram stain reference method, although performed similarly to culture.

We found higher rates of BV and CV using the investigational Aptima assays than our current methods, and these rates were in keeping with other studies and global

**TABLE 4** Performance of the Aptima investigational assays against reference methods

| Target | Sensitivity | Specificity |
|---|---|---|
| BV[a] | 77/79, 97.5% | 154/160, 96.3% |
| CV |  |  |
| – vs consensus Gram stain[b] | 35/35, 100% | 197/236, 83.5% |
| – vs Culture[c] | 78/81, 96.3% | 196/215, 91.2% |
| TV | 3/3, 100% | 294/294, 100% |

[a]Samples with an indeterminate Nugent score were excluded.
[b]Samples with yeast colonization were included as CV negative.
[c]Samples with scanty growth of yeast were included as culture negative.

**TABLE 5** Concordance of CV Gram stain reading

| First read | Second read | | | |
|---|---|---|---|---|
| | Negative | Yeast colonization | Yeast infection | Total discordant |
| Negative | 234 | 6 | 7 | 13 (5.3%) |
| Yeast colonization | 2 | 4 | 5 | 7 (63.6%) |
| Yeast infection | 3 | 1 | 35 | 4 (10.3%) |
| Total discordant | 5 | 7 | 12 | 24 (8.1%) |

estimates of prevalence (4, 5). Lillis *et al.* performed a clinical evaluation of over 1,400 clinician-collected and self-collected swabs performed on the Xpert Xpress MVP test and reported BV in 25.7%, CV in 19.4%, and TV in 1.0% by their comparator methods (21).

We found the Aptima BV assay performed very well against our reference method with a sensitivity and specificity of 97.5% and 96.3%, respectively. Schwebke *et al.* enrolled 1,519 patients to perform a comparison of the Aptima BV assay against a consensus Nugent score and modified Amsel criteria for both clinician and self-collected swabs (1). They detected BV in 49.5% by reference methods and reported similar performance to our sensitivities of 95% and 97.3% for clinician-collect and self-collect samples, respectively, but the specificities were lower (89.6% and 85.8%). Their study was different from ours in that six vaginal swab samples were collected from enrolled patients, and the reference method included the modified Amsel criteria (comprising pH, the presence of clue cells, and whiff test) (1). Ruffier d'Epenoux *et al.* evaluated the Aptima BV assay using 189 samples from inpatients in a French hospital and found BV in 13% of samples by reference Gram stain with Nugent score (22). They excluded samples with BV indeterminate Nugent score, similar to our study, and reported a sensitivity of 91.1% and a specificity of 94.4% (22). Interestingly, they noted that within the excluded BV indeterminate group, the higher the Nugent score, the greater the likelihood of a positive Aptima BV result: 70%, 50%, and 16.7% of samples with a Nugent score of 6, 5, and 4, respectively, were Aptima BV positive. Another study by Caza *et al.* compared 422 specimens but included indeterminate BV Nugent scores with clue cells as positive for BV and those without clue cells as BV negative.(23) In that study, 66 samples with discordance between the Nugent score and BV Aptima result were tested on a second molecular platform using the Seegene Allplex VS assay. This approach yielded positive and negative percent agreements of 98.4% and 95.9%, respectively, against the consensus BV result. We did not include clue cells in our study since this is not routine in our laboratory.

It is difficult to compare our data with prior studies utilizing alternative BV molecular assays since targets may differ and testing algorithms of commercial assays are usually proprietary. Differences in assay design are likely to explain some of the differences seen between available BV molecular assays.Caza *et al.* reported errors in Gram stain Nugent interpretation in 25/42 (59.5%) where the Gram stain was available for review, similar to other studies (14, 23). We also found variability in BV Gram stain interpretation, with discordant results in 20% of samples. The Aptima BV assay potentially improves the consistency of reporting and previous published work demonstrates acceptable reproducibility (18), although we did not test assay precision in our study. Furthermore, removal of the BV indeterminate category may reduce diagnostic uncertainty since the clinical significance of this category remains unclear (10, 11).

Our audit demonstrated that 66% of patients with a BV-positive Gram stain received treatment, compared with 5% of BV negative and 25% if BV indeterminate. Most

**TABLE 6** Comparison of CV Gram stain and culture

| Consensus Gram read | Yeast culture | | | | | |
|---|---|---|---|---|---|---|
| | No growth | Scanty growth | Light growth | Moderate growth | Heavy growth | Total |
| Negative | 197 | 13 | 9 | 10 | 4 | 233 |
| Yeast colonization | 1 | 0 | 0 | 0 | 3 | 4 |
| Yeast infection | 0 | 0 | 0 | 4 | 31 | 35 |

prescriptions occurred within a few days of the sample date (data not shown), indicating that this was likely in response to the Gram stain report. In our study, 56.5% of indeterminate BV samples returned a positive result on the Aptima BV assay, which may lead to more women receiving BV treatment should this assay be implemented (since 75% of women with an indeterminate result using current methods do not receive treatment). In our context, processing 1,800 swabs per month and extrapolating from our study and audit findings, this equates to around 100 additional patients receiving BV treatment per month. However, further study is warranted to explore the clinical impact of such a change. Of note, up to 40% of asymptomatic women may be positive for Aptima BV (18), highlighting the need for ongoing diagnostic stewardship interventions.

Evaluation of the CV component of the Aptima CV/TV assay was more problematic than for BV since either Gram stain or culture can be used for CV diagnosis (5, 12, 13). CV assay performance was dependent on the comparator method, which we found to be better when compared against culture than Gram stain. Coverage of the assay for yeast species known to cause CV depends on local epidemiology but likely exceeds 90% (5, 12, 13), and in our hands, the Aptima CV assay detected more yeast than either of the reference methods. Schwebke *et al.* did not demonstrate as much of a difference (30.0% positive by reference method compared to 28.6% by Aptima) (1), which may be because they included any growth as culture positive, whereas we classified scanty growth (<10 colonies) as negative. Higher sensitivity for the detection of yeast using culture or molecular methods compared with Gram stain is to be expected, and we demonstrated this. Similar to our findings, Caza *et al.* identified that 16/242 (6.6%) culture-negative samples were positive by Aptima CV, which was then tested on a second molecular platform using the Allplex VS assay. Of these, 10 (62.5%) were positive on the second assay, which improved the negative percentage agreement from 89.0% compared with culture to 95.4% compared with the consensus result. Positive percent agreement was 100%, regardless of whether compared with culture alone or consensus results including the second molecular test (23). Although we did not determine the relative limit of detection (LoD) for the different CV methods, published data indicate this to range from 41 cfu/mL to 9416 cfu/mL for the Aptima CV targets, depending on the yeast species (19). Ultimately, neither culture nor molecular methods differentiate yeast colonization from infection, and previous work demonstrated that 29% of asymptomatic women tested with Aptima CV were positive (19). Based on our findings, including scanty growth in the negative CV category would seem reasonable, but we cannot confirm the best diagnostic approach for CV. Further clinical evaluation of the CV assay is warranted to assess for any increase in treatment of yeast colonization as an unintended consequence of assay implementation.

Comparing the TV component of the CV/TV assay with the reference Aptima *T. vaginalis* assay yielded three positive detections and 100% concordance. This result was as expected, as the target sequence is the same between assays.

Molecular methods incur analyzer and reagent costs, but this may be justified by improved consistency of reporting and efficiencies in workflow. We found this assay simple to perform using the fully automated Hologic Panther instrument with minimal hands-on time. An additional benefit would be simplifying to a single swab type for investigation of the main infectious causes of vaginitis symptoms and the long-term stability of the sample, which contains a stabilizing buffer.

There are several limitations to our study. It was performed in a single laboratory, and we cannot comment on performance for patients aged <12 or >60 years. We did not evaluate for differences in performance according to whether the sample was clinician-collected or self-collected. We did not perform yeast identification, since this is not part of our routine process unless the clinical details indicate the presence of persistent infection or failed treatment. We therefore cannot comment on the accuracy of the CV assay according to yeast species, including for *N. glabrata* due to the very low numbers of this organism detected in our cohort. However, patients with recalcitrant infection would typically require culture for identification and susceptibility testing. Published

data indicate high accuracy of detection across five yeast species (*C. albicans, N. glabrata, C. parapsilosis, C. tropicalis*, and *C. dubliniensis*), but other species would not be detected (19); culture may still be necessary if yeast infection was strongly suspected despite a negative Aptima CV result. Finally, we did not prospectively evaluate the clinical impact of the molecular method compared with the reference method, and further investigation is warranted.

In conclusion, the Aptima BV and CV/TV assays performed well and would be a suitable alternative to conventional testing with the potential for quality improvement in laboratory diagnosis for vaginitis. We show the potential to increase BV diagnoses and improve the reproducibility and consistency of reporting, which could lead to more women being treated for this dysbiosis condition. However, further clinical studies are required to determine the true clinical impact of implementation, which could likely be managed safely with careful communication about the relevant changes and prudent use of interpretive comments. The additional cost compared with traditional methods may be a barrier to implementation.

## ACKNOWLEDGMENTS

We would like to thank Hologic, who provided reagents for this study and reviewed the study design. Hologic had no role in performing the study, analyzing the results, or writing this manuscript.

## AUTHOR AFFILIATIONS

[1]Department of Microbiology, Awanui Labs Dunedin, Dunedin Hospital, Dunedin, New Zealand

[2]Department of Microbiology and Immunology, University of Otago, Dunedin, New Zealand

## AUTHOR ORCIDs

Juliet Elvy http://orcid.org/0009-0001-4482-6758

## AUTHOR CONTRIBUTIONS

Juliet Elvy, Conceptualization, Data curation, Formal analysis, Investigation, Methodology, Validation, Writing – original draft, Writing – review and editing | Katelyn Carter, Conceptualization, Data curation, Formal analysis, Investigation, Methodology, Writing – review and editing | Jenna Paterson, Data curation, Formal analysis, Methodology, Writing – review and editing | Megan Smith, Conceptualization, Data curation, Methodology, Writing – review and editing | Gayleen Parslow, Conceptualization, Methodology, Writing – review and editing | James E. Ussher, Conceptualization, Data curation, Methodology, Writing – review and editing

## ADDITIONAL FILES

The following material is available online.

Open Peer Review

**PEER REVIEW HISTORY (review-history.pdf).** An accounting of the reviewer comments and feedback.

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
