## [Reviewer comments · Microbiology Spectrum]

Microbiology Spectrum

Clinical evaluation of the Hologic Panther Aptima BV and CV/TV assays for the diagnosis of vaginitis in Dunedin, Aotearoa New Zealand.

Juliet Elvy, James Ussher, Katelyn Carter, Jenna Paterson, Megan Smith, and Gayleen Parslow

Corresponding Author(s): Juliet Elvy, Awanui Labs New Zealand

Review Timeline:

Submission Date:	May 24, 2024
Editorial Decision:	July 2, 2024
Revision Received:	July 25, 2024
Editorial Decision:	September 3, 2024
Revision Received:	September 13, 2024
Accepted:	October 18, 2024

Editor: Ana Cabrera

Reviewer(s): Disclosure of reviewer identity is with reference to reviewer comments included in decision letter(s). The following individuals involved in review of your submission have agreed to reveal their identity: Megan Amerson-Brown (Reviewer #1)

Transaction Report:

DOI: <https://doi.org/10.1128/spectrum.01274-24>

Re: Spectrum01274-24 (Clinical evaluation of the Hologic Panther Aptima BV and CV/TV assays for the diagnosis of vaginitis in Dunedin, Aotearoa New Zealand.)

Dear Dr. Juliet Alexandra Elvy:

Thank you for the privilege of reviewing your work. Below you will find my comments, instructions from the Spectrum editorial office, and the reviewer comments.

Revision Guidelines

Sincerely,
Ana Cabrera
Editor
Microbiology Spectrum

Reviewer #1 (Comments for the Author):

This manuscript describes the performance of the Hologic Aptima BV and CV/TV diagnostic assay in a New Zealand hospital system when compared to current diagnostic methods that are used.

Major comments

1. Please use CV to represent candidiasis through out the document. It is confusing that paper has both CV and VVC used throughout.

2. Please refer to the Gram stain when discussing BV as Gram stain with Nugent scoring. Gram stain is ok to use when discussing candidiasis.
3. More should be added to the discussion as to why the performance for the samples with indeterminate Nugent score was so variable. Is this because of the targets that are on the panel for BV? What targets are present and how do they correspond with what we know about the BV microbiome?
4. It is mentioned that Nugent scoring for BV was discordant on 20% of repeat/duplicate samples. Since this is the case, why was Nugent scoring instead of Amsel's criteria used as the reference method?
5. How were you able to distinguish the sensitivity/specificity for CV compared to culture if culture is only performed in select cases and is not the routine diagnostic method?
6. Line 141 states that patients were excluded if prescription data was not available. Wouldn't you expect the negative cohort to not have an antimicrobial prescription?
7. More discussion is needed regarding the pathogenesis of BV and why the Hologic assay performs different than other BV assays. There are different targets on each of these assays. Because we do not fully understand what organisms or quantity of organisms cause BV vs don't cause BV.
8. State whether any asymptomatic BV was identified with the Aptima assay compared to the Nugent scoring technique.

Minor comments

1. update *Atopobium vaginae* to the latest reclassification- *Fannyhessea vaginae*
2. line 37-38 should be rephrased for better wording
3. line 43 mentions specific species that cause candidiasis. Are all of these organisms on the assay? What is the prevalence of these different species the frequency that they cause vaginal candidiasis.
4. It should be noted since this is a molecular assay that the presence of any *Trichomonas* is pathogenic, where the other assays BV and CV can be found as non-pathogenic normal flora.
5. line 51. inaccurate is not the appropriate word. The sentence should indicate that it is difficult to differentiate the cause of vaginitis based on clinical symptoms alone.
6. line 53-54. Nugent scoring is not quantitative but rather a semiquantitative scoring system.
7. Does the indeterminate result for Nugent scoring signify anything? Paired with clinical symptoms is it used to treat? Please provide more information here regarding clinical practice and the usefulness of the indeterminate zone.
8. It should be noted that TV nucleic acid testing is the clinically recommended testing method over wet mounts or culture. It is very rare to find a lab that cultures TV
9. Please state if there are any difference in the TV primers on the TV, CT, NG assay compared to the CV/TV assay.
10. If no third read was done for discordant CV results how was a decision made to classify the sample?
11. Was culture performed on all yeast samples regardless of Gram stain results? If so this is discordant to what was previously stated in the manuscript
12. Line 150. What does an invalid Aptima result indicate?
13. Line 162. Is Gram stain and culture performed for CV on all isolates tested? This should be clarified. It is not clear in the methods.
14. Line 164. Why is CT mentioned here. If the manuscript is focusing on BV that data should indicate there were x number of samples that were positive for BV and CT but CT should not be singled out since it is not included on the test panel. Line 166 is the important data for this study.

Reviewer #2 (Comments for the Author):

My primary problem with the paper is my classical epidemiological need for consistent definition. The authors define bacterial vaginosis (BV) and vulvovaginal candidiasis (VVC) with somewhat loose definitions. I strongly suggest that they use the American CDC Sexually Transmitted Infections Treatment Guidelines. VVC definition is multifactorial with more than just observation of pseudohyphae on a slide and positive culture. Furthermore, culture alone is not indicative of VVC because a significant number of women have commensal yeast. If the authors have the ability, they need to get patient symptoms and clinician observed signs to further define VVC. There are papers in the literature that discuss VVC and BV and its definitions.

The use of Amies swabs as the specimen transport methodology needs to be referenced or at least validation information presented. There are many formats in which this swab is available: dry, with inoculate solution, or agar gel. The swab type and transport format need to be defined and presented. Amies swabs can allow for the overgrowth of yeast if not handled properly, which could have influenced the author's definitions for positive VVC.

The conclusion that the Hologic BV assay appeared to be almost evenly split into the intermediate Nugent scores is not surprising given that the intermediate state is the 'top of the hill,' so to speak. If this is observed after reanalysis, it would be interesting to see if the split can be correlated with a specific point in the intermediate BV score range.

Reviewer #1 (Comments for the Author):

This manuscript describes the performance of the Hologic Aptima BV and CV/TV diagnostic assay in a New Zealand hospital system when compared to current diagnostic methods that are used.

Major comments

1. Please use CV to represent candidiasis through out the document. It is confusing that paper has both CV and VVC used throughout.

We thank the reviewer for picking up this inconsistency. We have changed the manuscript to use CV throughout.

2. Please refer to the Gram stain when discussing BV as Gram stain with Nugent scoring. Gram stain is ok to use when discussing candidiasis.

We have amended the manuscript to incorporate this suggestion.

3. More should be added to the discussion as to why the performance for the samples with indeterminant Nugent score was so variable. Is this because of the targets that are on the panel for BV? What targets are present and how do they correspond with what we know about the BV microbiome?

The BV targets included in the assay are noted in the material and methods section. We have added the following to the first paragraph of the discussion:

"This may represent the limitation of the Nugent score for classifying women as having BV or not if microscopic changes in the microbiome are less clear cut. Alternatively, the Aptima BV assay may either be missing or overcalling BV in these samples due to the targets included (or not included) in the assay. There are a growing number of genera implicated in the BV disease process and which are not detected by the assay (e.g. *Mobiluncus*, *Prevotella*, *Bacteroides*);¹⁹ future studies could address this problem by testing indeterminant samples with an additional PCR assay that includes more targets, or by metagenomic assessment."

4. It is mentioned that Nugent scoring for BV was discordant on 20% of repeat/duplicate samples. Since this is the case, why was Nugent scoring instead of Amsel's criteria used as the reference method?

Nugent scoring was used as this is the standard method in place in our laboratory. Amsel's criteria includes clinical criteria which we are not able to apply.

5. How were you able to distinguish the sensitivity/specificity for CV compared to culture if culture is only performed in select cases and is not the routine diagnostic method?

Culture was performed in all cases for the purposes of this study. This has been clarified in the method.

6. Line 141 states that patients were excluded if prescription data was not available. Wouldn't you expect the negative cohort to not have an antimicrobial prescription?

This refers to the patient data not being available on the portal, not that there was no prescription for the antibiotic. We have clarified this in the manuscript.

7. More discussion is needed regarding the pathogenesis of BV and why the Hologic assay performs different than other BV assays. There are different targets on each of these assays. Because we do not fully understand what organisms or quantity of organisms cause BV vs dont cause BV.

We have added the following to the discussion, lines 262-265:

It is difficult to compare our data with prior studies utilising alternative BV molecular assays since targets may differ and testing algorithms of commercial assays are usually proprietary. Differences in assay design are likely to explain some of the differences seen between available BV molecular assays.

8. State whether any asymptomatic BV was identified with the Aptima assay compared to the Nugent scoring technique.

We do not perform testing if asymptomatic as part of our standard operating procedure: testing is only performed if the clinical request form indicates that the patient is symptomatic. We have clarified this in the Methods section (line 90).

Minor comments

1. update Atopobium vaginae to the latest reclassification- Fannyhessea vaginae We have amended as suggested.

2. line 37-38 should be rephrased for better wording

The wording has been changed to: "The prevalence of BV is estimated to be up to 25% globally, but this may vary according to demographic and patient factors.³ The prevalence of BV in Aotearoa New Zealand (NZ) has not been documented."

3. line 43 mentions specific species that cause candidiasis. Are all of these organisms on the assay? What is the prevalence of these different species the frequency that they cause vaginal candidiasis.

The targets included in the assay (*C. albicans*, *C. tropicalis*, *C. parapsilosis*, and *C. dubliniensis*, *N. glabrata*) are noted in the methods section (lines 131-132). Mention of *Pichia* has been removed from line 43.

Line 40 has been edited and referenced: CV is caused by yeast infection, predominantly with *Candida albicans* (80-90% of cases).⁴

In addition, lines 287-289 in the discussion have been changed to: Coverage of the assay for yeast species known to cause CV depends on local epidemiology but likely exceeds 90%^{5,12,13} and, in our hands, the Aptima CV assay detected more yeast than either of the reference methods.

4. It should be noted since this is a molecular assay that the presence of any *Trichomonas* is pathogenic, where the other assays BV and CV can be found as non-pathogenic normal flora.

We already mention that yeasts are part of the normal vaginal flora (line 40) and colonisation may be commonly detected (lines 301 to 303). We also state that BV may be detected in up to 40% of asymptomatic women (line 282). We have amended line 36-7 as follows: Symptoms of BV include offensive vaginal discharge, although up to 50% may be asymptomatic.³

5. line 51. inaccurate is not the appropriate word. The sentence should indicate that it is difficult to differentiate the cause of vaginitis based on clinical symptoms alone.

We have changed "inaccurate" to "difficult".

6. line 53-54. Nugent scoring is not quantitative but rather a semiquantitative scoring system.

We have made this change.

7. Does the indeterminate result for Nugent scoring signify anything? Paired with clinical symptoms is it used to treat? Please provide more information here regarding clinical practice and the usefulness of the indeterminate zone.

As discussed above, there is likely to be a spectrum from normal vaginal flora to clear dysbiosis. An indeterminate Nugent result means that the Gram stain appearance can not be classified as either normal or clearly dysbiotic. Little is known regarding clinical practice in the setting of clinical symptoms and an indeterminate result. As found in our study, treatment was less common with an indeterminate result than with a positive Nugent score. We have added to lines 57-58 that the clinician must decide if treatment is warranted.

8. It should be noted that TV nucleic acid testing is the clinically recommended

testing method over wet mounts or culture. It is very rare to find a lab that cultures TV

We have added that TV NAAT has now become the recommended diagnostic method available (line 61-62).

9. Please state if there are any difference in the TV primers on the TV, CT, NG assay compared to the CV/TV assay.

As this is a commercial assay, it is proprietary information. Therefore it is not possible to add this information.

10. If no third read was done for discordant CV results how was a decision made to classify the sample?

For clinical reporting, the initial result was reported to the clinician. Discordant samples were excluded from analysis which has been clarified in the methods section.

11. Was culture performed on all yeast samples regardless of Gram stain results? If so this is discordant to what was previously stated in the manuscript

Samples in the study all had culture performed, although the current procedure in place in the laboratory is for selected culture. We have removed the comment about selective culture in the introduction to avoid confusion.

12. Line 150. What does an invalid Aptima result indicate? A failed result – added (failed) beside invalid for clarity

13. Line 162. Is Gram stain and culture performed for CV on all isolates tested? This should be clarified. It is not clear in the methods.

Yes. We have modified the Methods section to improve clarity.

14. Line 164. Why is CT mentioned here. If the manuscript is focusing on BV that data should indicate there were x number of samples that were positive for BV and CT but CT should not be singled out since it is not included on the test panel. Line 166 is the important data for this study.

We have removed line 164.

Reviewer #2 (Comments for the Author):

My primary problem with the paper is my classical epidemiological need for consistent definition. The authors define bacterial vaginosis (BV) and vulvovaginal candidiasis (VVC) with somewhat loose definitions. I strongly suggest that they use the American CDC Sexually Transmitted Infections Treatment Guidelines. VVC definition is multifactorial with more than just observation of pseudohyphae on a slide and positive culture.

Furthermore, culture alone is not indicative of VVC because a significant number of women have commensal yeast. If the authors have the ability, they need to get patient symptoms and clinician observed signs to further define VVC. There are papers in the literature that discuss VVC and BV and its definitions.

We acknowledge this but this is not a clinical study and we do not have access to patients' clinical records. However, it is important to note that we only perform testing on symptomatic patients (or at least we rely on the clinician giving us clinical information to indicate symptoms are present). This has been clarified in the methods section.

The use of Amies swabs as the specimen transport methodology needs to be referenced or at least validation information presented. There are many formats in which this swab is available: dry, with inoculate solution, or agar gel. The swab type and transport format need to be defined and presented. Amies swabs can allow for the overgrowth of yeast if not handled properly, which could have influenced the author's definitions for positive VVC.

We have clarified in the Methods section that Amies agar gel was used.

The conclusion that the Hologic BV assay appeared to be almost evenly split into the intermediate Nugent scores is not surprising given that the intermediate state is the 'top of the hill,' so to speak. If this is observed after reanalysis, it would be interesting to see if the split can be correlated with a specific point in the intermediate BV score range.

The score was not recorded, only the categorical result. However, given the differences seen in categorical scoring between scientists, it is likely that scores are less reproducible.

Re: Spectrum01274-24R1 (Clinical evaluation of the Hologic Panther Aptima BV and CV/TV assays for the diagnosis of vaginitis in Dunedin, Aotearoa New Zealand.)

Dear Dr. Juliet Alexandra Elvy:

Thank you for the privilege of reviewing your work. Below you will find my comments, instructions from the Spectrum editorial office, and the reviewer comments.

Revision Guidelines

Sincerely,
Ana Cabrera
Editor
Microbiology Spectrum

Reviewer #1 (Comments for the Author):

no additional comments.

Reviewer #2 (Comments for the Author):

Thank you for addressing my concerns.

Just to signify that this study is from a laboratory, modify the title to 'Clinical **laboratory** evaluation . . .'

Around Lines 98-99, What is the typical time from collection to clinic of an Amies swab? And what is the prescribed travel temperature?

Line 126, change "in O₂" to 'aerobically'.

Lines 235 and 236, The potential interpretation that since the intermediates are split between negative and positive Aptima assay results is a limitation of the Nugent score is inverted logic. It is important to remember that the Nugent system was designed against clinical and culture observations. The much later development of the Aptima assay for BV needed a clear positive. Converting a quantitative result to a qualitative one requires a formal assigned cutoff. Hologic's quantitative cutoff appears to be liberal enough to ensure that all traditional Nugent positive scores are included, even if it has to dip into the intermediate scores-which many clinicians will still treat if clinical signs are present. The sentence should be removed or clarified and expanded upon, especially since the authors are thinking along this track in the subsequent paragraph.

Line 264, after "Nugent score of 6, 5, and 4" put '(Intermediate Nugent scores, whereby the higher the score the more likely the subject has BV)'-or something just as definitive.

Second Response to Reviewers 13.9.2024

Reviewer #1 (Comments for the Author):

no additional comments. **No response required.**

Reviewer #2 (Comments for the Author):

Just to signify that this study is from a laboratory, modify the title to 'Clinical **laboratory** evaluation . . .'. **The title has been amended as requested**

Around Lines 98-99, What is the typical time from collection to clinic of an Amies swab? And what is the prescribed travel temperature? **Swabs are transported on the next available transport courier, usually the same day, or refrigerated overnight for transport the following day. Swabs are placed in an insulated chilly bin along with other diagnostic samples.**

Line 126, change "in O₂" to 'aerobically'. **This has been amended as requested.**

Lines 235 and 236, The potential interpretation that since the intermediates are split between negative and positive Aptima assay results is a limitation of the Nugent score is inverted logic. It is important to remember that the Nugent system was designed against clinical and culture observations. The much later development of the Aptima assay for BV needed a clear positive. Converting a quantitative result to a qualitative one requires a formal assigned cutoff. Hologic's quantitative cutoff appears to be liberal enough to ensure that all traditional Nugent positive scores are included, even if it has to dip into the intermediate scores-which many clinicians will still treat if clinical signs are present. The sentence should be removed or clarified and expanded upon, especially since the authors are thinking along this track in the subsequent paragraph.

We have removed this sentence:

This may represent the limitation of the Nugent score for classifying women as having BV or not if microscopic changes in the microbiome are less clear cut.

And edited the next sentence to read as follows:

There are a growing number of genera implicated in the BV disease process which are not detected by the Aptima assay (e.g. *Mobiluncus*, *Prevotella*, *Bacteroides*);²⁰ future studies could investigate this further by testing indeterminate BV samples with PCR assays which include more targets, or by metagenomic assessment.

Line 264, after "Nugent score of 6, 5, and 4" put '(Intermediate Nugent scores, whereby the higher the score the more likely the subject has BV)'-or something just as definitive.

This sentence already includes such a statement:

Interestingly, they noted that, within the excluded BV indeterminate group, the higher the Nugent score, the greater the likelihood of a positive Aptima BV result: 70%, 50%, and 16.7% of samples with a Nugent score of 6, 5, and 4, respectively, were Aptima BV positive.

Re: Spectrum01274-24R2 (Clinical evaluation of the Hologic Panther Aptima BV and CV/TV assays for the diagnosis of vaginitis in Dunedin, Aotearoa New Zealand.)

Dear Dr. Juliet Alexandra Elvy:

Your manuscript has been accepted, and I am forwarding it to the ASM production staff for publication. Your paper will first be checked to make sure all elements meet the technical requirements. ASM staff will contact you if anything needs to be revised before copyediting and production can begin. Otherwise, you will be notified when your proofs are ready to be viewed.

Sincerely,
Ana Cabrera
Editor
Microbiology Spectrum